# The Efficacy of Exercise Training for Cutaneous Microvascular Reactivity in the Foot in People with Diabetes and Obesity: Secondary Analyses from a Randomized Controlled Trial

**DOI:** 10.3390/jcm11175018

**Published:** 2022-08-26

**Authors:** Sean Lanting, Kimberley Way, Angelo Sabag, Rachelle Sultana, James Gerofi, Nathan Johnson, Michael Baker, Shelley Keating, Ian Caterson, Stephen Twigg, Vivienne Chuter

**Affiliations:** 1Faculty of Health and Medicine, School of Health Sciences, University of Newcastle, Ourimbah, NSW 2258, Australia; 2Faculty of Health and Medicine, Discipline of Exercise and Sports Science, University of Sydney, Camperdown, NSW 2006, Australia; 3The Boden Collaboration for Obesity, Nutrition, Exercise and Eating Disorders, University of Sydney, Camperdown, NSW 2006, Australia; 4The Charles Perkins Centre, University of Sydney, Camperdown, NSW 2006, Australia; 5School of Exercise Science, Australian Catholic University, Strathfield, NSW 2135, Australia; 6Centre for Research on Exercise, Physical Activity and Health, School of Human Movement and Nutrition Sciences, The University of Queensland, St Lucia, QLD 4072, Australia; 7School of Medicine, University of Sydney, Camperdown, NSW 2006, Australia; 8School of Health Sciences, Western Sydney University, Campbelltown, NSW 2751, Australia

**Keywords:** cutaneous blood flow, exercise training, laser-Doppler, microvascular reactivity, peripheral artery disease, post-occlusive reactive hyperaemia

## Abstract

It is unclear if cutaneous microvascular dysfunction associated with diabetes and obesity can be ameliorated with exercise. We investigated the effect of 12-weeks of exercise training on cutaneous microvascular reactivity in the foot. Thirty-three inactive adults with type 2 diabetes and obesity (55% male, 56.1 ± 7.9 years, BMI: 35.8 ± 5, diabetes duration: 7.9 ± 6.3 years) were randomly allocated to 12-weeks of either (i) moderate-intensity continuous training [50–60% peak oxygen consumption (VO_2peak_), 30–45 min, 3 d/week], (ii) low-volume high-intensity interval training (90% VO_2peak_, 1–4 min, 3 d/week) or (iii) sham exercise placebo. Post-occlusive reactive hyperaemia at the hallux was determined by laser-Doppler fluxmetry. Though time to peak flux post-occlusion almost halved following moderate intensity exercise, no outcome measure reached statistical significance (*p* > 0.05). These secondary findings from a randomised controlled trial are the first data reporting the effect of exercise interventions on cutaneous microvascular reactivity in the foot in people with diabetes. A period of 12 weeks of moderate-intensity or low-volume high-intensity exercise may not be enough to elicit functional improvements in foot microvascular reactivity in adults with type 2 diabetes and obesity. Larger, sufficiently powered, prospective studies are necessary to determine if additional weight loss and/or higher exercise volume is required.

## 1. Introduction

Diabetes is estimated to affect almost 500 million people globally [1]. Of the USD 116 billion attributed to treating diabetes and its complications in the United States, approximately 33% is linked to treatment of diabetes-related foot disease (DFD), which includes foot ulceration and amputation [2]. A combination of lower limb diabetes-related vascular pathologies including peripheral artery disease (PAD) and peripheral neuropathy are known to contribute to the development of DFD [2]. In addition, cutaneous microvascular disease has been proposed to reduce healing capacity through structural and functional changes to the microvasculature [3].

Diabetes-related microvascular disease is known to occur as a result of changes to nerve-axon reflex-related vasodilation [4], reduction in the production and effect of endothelial-derived vasodilators [5], and angiopathic structural changes, including reduced capillary lumen diameter and capillary density, and increased basement membrane thickening [6]. These changes disrupt tissue viability and stymie the timely process of wound healing [4]. In addition, diabetes-related autonomic neuropathy is suggested to lead to dysregulation of arteriovenous anastomoses, resulting in loss of sympathetic tone and relaxation of the shunts which divert blood away from the skin and is proposed to contribute to localised tissue hypoxia (capillary steal syndrome) [7].

In people with diabetes, the presence of obesity may further compromise the microvasculature and the functional capacity for tissue repair [8]. Known obesity-specific contributors to microvascular dysfunction include oxidative stress, capillary rarefaction, endothelial dysfunction, and pro-inflammatory adipokines and cytokines [9], as well as depletion in macrophage-dependent hydrogen sulfide [10]. These pathogenic changes can also contribute to a loss of cutaneous microvascular reactivity resulting in an insufficient hyperaemic inflammatory response to trauma and substantial impairment to the healing process [11,12].

Exercise has shown promise as a potential therapy for cutaneous microvascular disease [13]. Posited benefits of exercise on the vasculature are proposed to occur as a result of a direct effect on endothelial function. Specifically, increased repetitive vascular shear stress from exercise is suggested to directly result in increases in the bioavailability of nitric oxide [14]. Increase in nitric oxide improves vasodilatory capacity of the microvasculature [15] and exerts an anti-proliferative, anti-thrombotic and anti-inflammatory effect on the endothelium, as such, protecting against endothelial disease [16]. We have recently demonstrated that, in a previously inactive, adult population, moderate-intensity continuous aerobic training (MICT) had a moderate significant beneficial effect on cutaneous microvascular reactivity (effect size = 0.43, 95% CI: 0.08 to 0.78, *p* = 0.015) [13]. Several studies in diabetes cohorts have shown transient increases in cutaneous microvascular reactivity in response to local heating of the dorsum of the foot following an acute bout of MICT [17] and in those self-reporting MICT versus inactive individuals [18]. However, to date, clinical trials have not assessed lower limb outcomes and the results of clinical trials in upper limbs are inconsistent [19,20].

In people with obesity and type 2 diabetes, barriers to exercise, such as lack of time, physical discomfort, and boredom during exercise [21] may be somewhat mitigated by high-intensity interval training (HIIT). HIIT has previously demonstrated a wide range of cardiometabolic adaptations, including compared to MICT, such as decreased systolic blood pressure and oxidative stress, as well as increased high-density lipoproteins and availability of nitric oxide [22]. However, further investigation to determine if there is a measurable effect of either type of exercise training (HIIT or MICT) on cutaneous microvascular function in the lower limb in this population is required.

The aim of this study was to investigate the effect of 12 weeks of HIIT and MICT vs. sham exercise placebo (PLA) on cutaneous microvascular reactivity in the foot in previously inactive adults with type 2 diabetes and obesity. Secondarily, we investigated the effect of these interventions on small and large peripheral artery function. We hypothesised that MICT and HIIT would improve cutaneous microvascular reactivity and would improve small and large artery function in the presence of detectable peripheral arterial disease compared to PLA.

## 2. Materials and Methods

### 2.1. Design

This study comprised a secondary analysis of data collected in a trial investigating the effect of exercise interventions on cardiometabolic outcomes [23] that was prospectively registered with the Australian and New Zealand Clinical Trials Registry (ACTRN12614001220651). A three-arm, randomised, single-blind, controlled, parallel-group study was conducted at the Charles Perkins Centre, University of Sydney, Australia. Ethics approval was obtained from the University of Sydney Human Research Ethics Committee and the Royal Prince Alfred Research Ethics Committee Governance Office and conformed to the ethical guidelines of the 1975 Declaration of Helsinki. Following attainment of informed, written consent and determination of participant eligibility, peripheral vascular and anthropometric measurements were undertaken at the same location. All participants underwent baseline measurements as well as identical measures post-trial, at least 72 h following the final intervention session after 12 weeks. Measurement of post-occlusive reactive hyperaemia (PORH), as well as brachial, ankle and toe pressures and peripheral neuropathic status were obtained by the same podiatrist (SL) who has several years clinical and research experience and was blinded to group allocation. All measures were obtained in the morning, in a temperature-controlled room (22–24 °C), following at least 10 min of supine rest and acclimatisation. All participants were required to be in a fasted state (>10 h), were requested not to take their usual prescription medication on that morning until measurements had been undertaken, and to avoid physical exertion, caffeine, smoking and alcohol for at least two hours prior to peripheral vascular measurement.

Following all baseline measures, participants were randomly allocated to one of two experimental groups (HIIT or MICT), or to PLA, by means of an equally distributed pre-generated list of permuted blocks (www.randomization.com, accessed on 1 June 2015) held by independent researchers (NJ, SK), concealed by opaque envelopes and given to the researchers supervising the interventions (KW, AS, RS). Participants allocated to either the HIIT or MICT groups underwent an exercise stress test to ensure that their health would not be put at risk by participating in the intervention.

### 2.2. Participants

People with type 2 diabetes were recruited voluntarily via either poster advertisement or following telephone contact after identification in a recruitment database. Specifically, posters advertising the trial were displayed at the Charles Perkins Centre and in the endocrinology department of the nearby Royal Prince Alfred Hospital. Three trial researchers (KW, AS, RS) conducted telephone screening between June 2015 and February 2019 to determine volunteer suitability. Eligible participants were 18 y or older with a diagnosis of type 2 diabetes, BMI between 30–45 kg/m^2^ and inactive (<3 d or <150 min of physical activity per week by self-report). Volunteers were excluded if they were unable to lay supine for 40 min or had a contraindication to either brachial or toe systolic pressure measurements, such as mastectomy, history of vasospastic disorder or bilateral hallux injury or amputation. Due to the physical demands of the exercise assessments and interventions, exclusion criteria also included uncontrolled angina, cardiovascular disease, arrhythmias or blood glucose levels. Additional exclusion criteria included the inability to complete an MRI scan (claustrophobia or BMI > 45 kg/m^2^) or inability to attend the facility 3 d per week for 12 weeks for supervised interventions. To ensure participant suitability, a thorough physical examination and health-screen was undertaken by a medical practitioner prior to baseline measurements. Participant age and sex were recorded, height, weight, hip and waist circumference measured, and BMI calculated. Diabetes duration, medications, and history of hypertension were extracted from medical records.

### 2.3. Cutaneous Microvascular Reactivity

Cutaneous microvascular reactivity at the hallux was measured by PORH as determined by laser-Doppler fluxmetry. An automated MoorVMS-PRES unit with toe cuff (Moor Instruments Ltd., Axminster, UK) was applied to the right hallux pulp with the moorVMS-LDF2 laser-Doppler probe embedded in a VHP2 skin heat probe (Moor Instruments Ltd., Axminster, UK) in order to maintain local skin temperature at 33 °C. In brief, the 9 min protocol of continuous flux measurement from which outcome variables were extrapolated was 3 min resting flux, 3 min occlusion at 220 mmHg, and 3 min post-occlusion. Specific PORH variables considered for analysis were the area under the curve (AuC) index one min post-occlusion to one min pre-occlusion (AuC index), peak as a percentage of baseline flux (P%BL), peak flux (Peak) and time to peak flux (sec) post-occlusion (TtPeak). These variables were chosen due to their ability to depict the extent of response and response over time in the context of their previously demonstrated acceptable reliability [24]. MoorVMS analysis software Version 3.1 (Moor Instruments Ltd.) was used to analyse the PORH data.

### 2.4. Peripheral Arterial Measures

Systolic pressure measures were attained at brachial, ankle and toe level in order to calculate bilateral ABI and TBI. An automatic blood pressure monitor, BP A100-30 (microlife^®^, Widnau, Switzerland), was used to measure systolic brachial pressure in right and left arms, with the cuff placed 2–3 cm from the inner fold of the elbow. Bilateral systolic ankle pressures were attained using an ERKA^®^ aneroid sphygmomanometer (Kallmeyer Medizintechnik GmbH & Co., Ltd., Bad Tölz, Germany), 8 Hz Bidop ES-100V3 hand-held Doppler^®^ (Hadeco, Kawasaki, Japan), and Aquasonic^®^ ultrasound transmission gel (Parker Laboratories, Fairfield, NJ, USA), in combination with, as required, an adult standard, adult large or adult extra-large inflatable cuff (Liberty Health Care^®^, Ashmore, London, UK) [25]. The pressure cuff was placed around the lower one third of the tibia, proximal to the malleoli and for each pressure measurement the cuff was inflated 20–30 mmHg higher than the last audible Doppler signal and then deflated to the point where the arterial waveform and audible signal returned. The ABI was calculated using the highest of the posterior or anterior tibial artery pressures from each leg over the higher of the two brachial pressures, in accordance with current guidelines [25]. Toe pressures were measured using a 2.5 cm Kami-Hadeco^®^ inflatable digital cuff (Hadeco, Kawasaki, Japan), a PG-21^®^ PPG probe (Hadeco, Kawasaki, Japan) and an ERKA^®^ aneroid sphygmomanometer (Kallmeyer Medizintechnik GmbH & Co., Ltd., Bad Tölz, Germany). All pressure gauges used in this study were newly calibrated. The toe pressure was divided by the highest systolic brachial pressure out of the left and right arm to calculate the TBI.

### 2.5. Anthropometric Measures, Cardiorespiratory Fitness and Neuropathic Status

Prior to measurement of primary and secondary outcomes, anthropometric measures were conducted by the clinicians overseeing the interventions (KW, AS and RS). A stadiometer (Tanita HR-200 Wall Mounted Height Rod, Arlington Heights, IL, USA) was used to measure stature and a digital platform scale used to measure body weight (Tanita BC-418 Body Composition Analyzer, Tanita Corporation, Tokyo, Japan). Waist circumference was measured in triplicate at the horizontal plane, midway between the inferior margin of the ribs and the superior border of the iliac crest at end-expiration, and hip circumference was measured at the widest point of the buttocks. Measurement of cardiorespiratory fitness have been described previously [23,26], though, briefly, breath-by-breath analysis was collected (Ultima PFX pulmonary function/stress testing system, MGC Diagnostics, Saint Paul, MN, USA) during a graded maximal exercise test to measure peak oxygen consumption (VO_2peak_). All tests incorporated a 3-min warm-up at 35 W for women and 65 W for men. The workloads were incrementally adjusted by 25 W every 150 s until volitional fatigue. Weight (m^2^) was divided by height to calculate the BMI. Peripheral neuropathic status was determined by use of a neurothesiometer (Wilford Industrial, Nottingham, UK) and 10 g 5.07 Semmes–Weinstein monofilament (North Coast Medical, Morgan Hill, CA, USA), and performed by the clinician undertaking the primary outcome measures (SL). Neuropathic status was classified as abnormal if both neurological tests were positive for a deficit [27].

### 2.6. Interventions

Participants randomly allocated to either MICT or HIIT were required to partake in three supervised sessions performed on an electronically braked cycle ergometer (Corival, Lode, Groningen, The Netherlands) per week for 12 weeks; however, up to 15 weeks to complete the 36 sessions were allowed if compliance was affected. Exercise sessions were supervised by an accredited exercise physiologist (KW, AS, RS). During each exercise session, blood pressure, heart rate (Polar, Polar FS1), rate of perceived exertion and symptoms of hypoglycaemia were monitored and recorded at regular intervals.

#### 2.6.1. High-Intensity, Low-Volume Interval Training (HIIT)

The HIIT sessions included a 10 min warm up at 50% VO_2peak_, followed by a 1 × 4 min high intensity bout designed to elicit 90% VO_2peak_, and a 5 min cool-down at 50% VO_2peak_.

#### 2.6.2. Moderate Intensity Continuous Training (MICT)

The MICT sessions incorporated a 5 min warm up and 5 min cool down at 50% VO_2peak_ either side of MICT. MICT progressed from 30 min at 50% VO_2peak_ in week 1 to 45 min at 60% VO_2peak_ by week 4 to enable participant compliance.

#### 2.6.3. Sham-Exercise Placebo (PLA)

Participants allocated to PLA attended once every two weeks for a session lasting ≤30 min. The supervised session involved performing static stretches of the major muscle groups of the legs, chest, arms and back followed by floor core exercises targeting transverse abdominis, gluteal and internal/external oblique muscle groups. To maintain familiarity with the cycle ergometer, 5 min of very low intensity (20 W) exercise on the cycle ergometer was undertaken as a warm-up and cool-down during each session.

### 2.7. Sample Size

As this study is a secondary analysis of a randomised control trial, the power calculation was not determined a priori. The power calculation was, thus, based on an effect size consistent with our recent systematic review and meta-analysis [13] of the effect of MICT on cutaneous microvascular reactivity (approx. 0.5) at a power of 80%, alpha of 0.05 and assuming an attrition rate of 20%, indicating 17 participants per group (*n* = 51) would be required to provide adequate power for the ANCOVA. However, 33 volunteers were recruited due to limitations in funding for the trial.

### 2.8. Statistical Analysis

Compliance to the exercise interventions was calculated as a percentage of those completed from the number of sessions available. Statistical analysis was undertaken using an intention-to-treat analysis, with group mean change scores imputed for dropouts and missing data [28]. Between-group differences at baseline for participant characteristics were calculated using one-way analysis of variance for continuous data and chi-squared test for categorical data. The primary analyses explored effects between the control and the exercise groups. The group effect of primary (AuC index, P%BL, Peak and TtPeak) and secondary (systolic toe pressure, TBI and ABI) outcome measures was assessed by analysis of covariance (ANCOVA), with LSD post hoc comparison used to locate significant differences between interventions and placebo. For analysis purposes, slower TtPeak, and lower P%BL, Peak and AuC index were considered directionally pathological [29,30,31].

## 3. Results

Ninety-seven individuals were contacted by phone to determine suitability for involvement in the trial. Forty-three of those interviewed did not meet the inclusion criteria, 13 declined participation and eight were excluded due to uncontrolled cardiovascular complications. Thirty-three people (18 male, 15 female) underwent baseline measures and randomisation, of which three were lost to follow-up citing lack of time, Figure 1. The remaining 30 participants (56.1 ± 7.9 years, duration of diabetes: 7.9 ± 6.3 years, BMI: 35.8 ± 5.0 kg/m^2^) completed the intervention period. Baseline participant characteristics and between group differences for control and intervention groups are detailed in Table 1. There were no statistically significant differences between groups at baseline for all outcome measures. There were no adverse events reported. All participants allocated to the exercise interventions completed the required 36 supervised exercise sessions in the allotted time, with compliance of completers, 91%, 97%, and 56% for MICT, HIIT and PLA, respectively. Reasons for poorer compliance in the PLA group were primarily due to the less frequent contact time or dissatisfaction with group allocation.

### Primary and Secondary Outcomes

There was no significant group effect for cutaneous microvascular reactivity, as measured by PORH variables, AuC index, P%BL, Peak or TtPeak (*p* > 0.05) (Table 2). Similarly, there was no statistically significant group effect for peripheral arterial measures (systolic toe pressure, TBI, or ABI), (*p* > 0.05). In addition, LSD post hoc tests did not identify any significant differences in primary or secondary outcome measures between either intervention group and the placebo group (*p* > 0.05). The effects of exercise interventions on cardiovascular, metabolic and other outcome measures (Table 3) have been extensively described previously [23,26].

## 4. Discussion

By randomised controlled trial, we investigated the effect of regular exercise training on cutaneous microvascular reactivity in the foot in people with type 2 diabetes and obesity. To the authors’ knowledge, this is the first RCT analysing the effect of exercise training on PORH outcome measures in the foot, or TtPeak and AuC index outcomes, in any anatomical region, in people with diabetes. The primary finding of this study was that, in an adult population with type 2 diabetes and obesity, 12 weeks of supervised exercise (MICT or HIIT) did not have a statistically significant effect on PORH measured at the hallux, compared to sham exercise placebo or to baseline measures (*p* > 0.05). Though this study is underpowered, these are the first data from RCT providing lower limb cutaneous microvascular measures, which is highly relevant given the overwhelming burden of diabetes-related foot disease. This data suggests that a higher volume of exercise and larger sample studies are important considerations for interventions involving physical activity. In addition, supervised exercise interventions did not have a significant effect on lower limb macrovascular measures (systolic toe pressure, TBI or ABI) in this cohort.

Previous studies have investigated the effect of MICT [19,20] and HIIT [20] interventions on cutaneous microvascular reactivity in the upper limb in people with type 2 diabetes with conflicting results. Our findings of no significant effect of MICT on cutaneous microvascular reactivity in the foot as determined by PORH is supported by a lack of effect of MICT in response to ACh iontophoresis and to local heating at the forearm [19]. In a group of 60 people with well-controlled type 2 diabetes, Middlebrooke et al., [19] found no effect of MICT (30 min, three times weekly at 70–80% of maximal heart rate for six months) on cutaneous microvascular reactivity. The consistency of these findings indicate that it is possible that the well-controlled nature of the disease and lack of historical foot complications in our participant population place them at lower likelihood of having significant microvascular pathology and, therefore, of demonstrating an effect of the intervention. However, we have recently demonstrated microvascular dysfunction in people with overweight and obesity with well-controlled diabetes [32].

We found no significant effect of HIIT on cutaneous microvascular reactivity. These results are in contrast to findings reported by Mitranun et al., who found a significant effect of HIIT (30–40 min session, three times per week for 12 weeks at 50–85% VO_2peak_) on cutaneous microvascular reactivity at the forearm [20]. While these conflicting findings may be related to lack of power of our sample to detect a medium-sized effect of MICT/HIIT on cutaneous microvascular reactivity, they may also be explained by differences in HIIT dosage. In our present study, we employed a 1 × 4 min high-intensity bout at 90% VO_2peak_, versus the 4–6 × 1 min bouts used by Mitranun et al. Both trials included gentle aerobic exercise as part of a warm-up and cool-down, but Mitranun et al. also used gentle aerobic exercise between HIIT bouts and, overall, their exercise intervention session was approximately twice as long as the one used in our study. Therefore, significant changes to cutaneous microvascular reactivity observed in their study may have been the result of repetitive, intermittent sheer stress [14], that was not elicited by our HIIT dosage. Differences in the duration of diabetes could partly explain the mixed findings regarding the effect of MICT and HIIT on cutaneous microvascular reactivity between our present study (mean duration of seven years) and prior studies, where the mean duration of disease ranged from four years [19] to 20 years [20]. In addition, the change in microvascular function reported by Mitranun et al. may be associated with concurrent reductions in measures of obesity and body fatness. Previous research has shown an association between obesity and both blunted reactive hyperaemic response and reduced endothelium-dependent vasodilation [33,34]. Further, we have recently shown that impaired cutaneous microvascular reactivity is related to higher abdominal subcutaneous adipose tissue and total percentage body fat in people with obesity and type 2 diabetes [32]. The implication is that weight loss is likely a primary trigger for improvements to cutaneous microvascular function regardless of the volume or intensity of exercise, and that this should be a goal of exercise therapy in this population. The more pronounced improvements observed by Mitranun et al., were in the HIIT group, which also corresponded to a significant increase in the bioavailability of nitric oxide. Therefore, future research should also aim to establish the required volume and intensity of HIIT necessary to precipitate both weight loss and improved endothelial function in people with type 2 diabetes and obesity.

The lack of effect of MICT or HIIT on larger artery function, as assessed by the ABI, are unsurprising given the absence of detectable PAD within the participant population. While there is some evidence to suggest that reductions in small arterial pressure in the foot occur in the presence of increased measures of body fatness [32], and there was a group mean systolic toe pressure change of +16.3 mmHg following MICT, these results did not reach clinical significance. However, this is worthy of investigation in populations with evidence of PAD and reduced systolic toe pressures as such a change could have clinical significance in a population at risk of ischaemic complications.

The results of this trial should be viewed in light of several limitations in addition to the underpowered sample size, as a result of exhausted funds, limiting our ability to detect significant change. Firstly, we cannot be certain about the number of participants who had existing cutaneous microvascular disease, as there is no reference standard to determine this, and PAD was determined by ABI, TBI and systolic toe pressure only. Therefore, lack of disease, or uneven distribution of those with disease between the groups may have reduced the magnitude of the effect of the exercise interventions on the primary and secondary outcomes. There was also a significant difference between the groups for VO_2peak_ (mL/kg/min) and this may have confounded findings, as may have the relatively low compliance of the PLA group. PORH is considered a mixed measure of both endothelial-dependent and -independent functions [30] and we are unable to comment on the specific nature of these responses. Additionally, the effect of resistance training on cutaneous microvascular reactivity was not explored in our study, and, although long-term benefits have previously shown promise [35], this remains inconclusive [36]. Lastly, our findings are generalizable to a community-based population of previously inactive adults with type 2 diabetes and obesity without evidence of diagnosed microvascular or peripheral arterial disease.

## 5. Conclusions

The results of this randomised control trial suggest that the efficacy of MICT and HIIT to improve cutaneous microvascular reactivity in the foot in people with diabetes and obesity remains unclear. Adequately powered prospective trials are required to confirm the effectiveness of exercise therapies on peripheral micro- and macrovascular function. Further, investigation is warranted to establish an effective dosage and intensity of exercise required to elicit significant changes and to determine if weight loss is required to trigger improved peripheral vascular function.

## Figures and Tables

**Figure 1 jcm-11-05018-f001:**
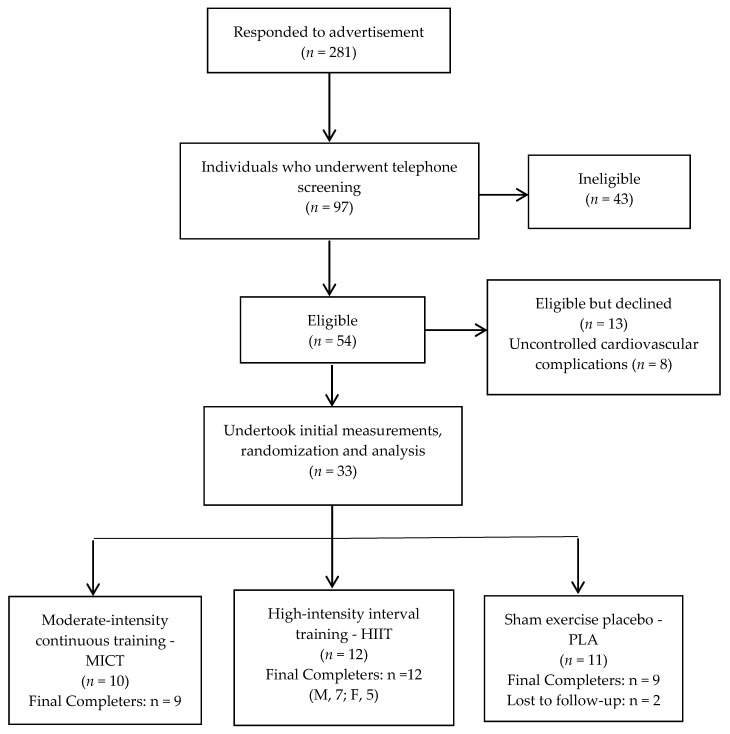
CONSORT flow diagram of participants through the trial.

**Table 1 jcm-11-05018-t001:** Characteristics of participants at baseline.

Characteristic	Moderate-Intesity Continuous Training—MICT (*n* = 10)	High-Intensity Interval Training—HIIT (*n* = 12)	Control Group Placebo—PLA (*n* = 11)	Between-Group (*p*-Value)
Age (year), mean (SD)	56.8 (6.8)	58.3 (6.9)	52.9 (8.7)	0.26
Sex, number male (%)	4 (40)	7 (58)	7 (64)	0.53
Weight (kg), mean (SD)	95.6 (16.9)	110.4 (14.5)	106.0 (16.9)	0.13
Height (m), mean (SD)	1.67 (0.1)	1.72 (0.09)	172 (0.09)	0.44
Body mass index (kg/m^2^)	33.9 (3.2)	37.5 (5.2)	35.8 (5.4)	0.26
Waist circumference (cm), mean (SD)	110.2 (9.7)	122.0 (11.3)	116.2 (13.1)	0.09
Hip circumference (cm), mean (SD)	118.4 (8.6)	123.4 (11.6)	118.7 (10.6)	0.49
Visceral adipose tissue (cm^3^), mean (SD)	6327 (2489)	8535 (2935)	7983 (2816)	0.21
Abdominal subcutaneous adipose tissue (cm^3^), mean (SD)	11,574 (2682)	11,810 (3203)	12,118 (3969)	0.94
Diabetes duration (year), mean (SD)	7.3 (5.3)	9.3 (7.0)	6.9 (5.9)	0.65
H_b_A1c [% National Glycohemoglobin Standardaization Program (NGSP) units], mean (SD)	7.2 (1.3)	7.1 (1.4)	7.6 (1.5)	0.63
Fasting blood glucose (mmol/L), mean (SD)	7.7 (2.0)	7.6 (2.7)	9.1 (3.9)	0.49
VO_2peak_ (mL/kg/min), mean (SD)	21.2 (5.3)	20.8 (2.5)	19.6 (4.3)	0.69
Resting heart rate (bpm), mean (SD)	69 (8)	68 (8)	74 (12)	0.33
Oral hypoglycaemic agents alone or in combination with insulin, number (%)	8 (80)	12 (100)	11 (100)	0.86
Anti-hypertensive therapy, number (%)	5 (50)	10 (83)	7 (64)	0.25
Lipid-lowering therapy, number (%)	8 (80)	5 (42)	7 (64)	0.06
Current smoker, number (%)	0 (0)	1(8)	0 (0)	0.41
Systolic blood pressure (mmHg), mean (SD)	129 (11.4)	142 (9.8)	137 (18.6)	0.12
Toe systolic pressure (mmHg), mean, (SD)	102 (21.0)	114 (20)	104.4 (29.6)	0.50
Ankle-brachial index, mean (SD)	1.08 (0.06)	1.03 (0.13)	1.06 (0.13)	0.60
Toe-brachial index, mean (SD)	0.79 (0.15)	0.80 (0.11)	0.76 (0.21)	0.87
Sensory neuropathy, number (%)	1 (10)	0 (0)	0 (0)	0.31
*Post-occlusive reactive hyperaemia*				
Baseline [FU (Flux Units)], mean (SD)	37.2 (35.1)	27.6 (21.6)	37.15 (21.15)	0.63
AuC, index (post FU:pre FU), mean (SD)	3.60 (2.33)	3.83 (2.44)	3.08 (0.96)	0.69
P%BL, mean (SD)	501.1 (200.0)	542.3 (307.9)	447.9 (154.1)	0.66
Peak (FU), mean (SD)	150.4 (117.2)	119.3 (79.2)	152.9 (87.2)	0.67
Time to peak (s), mean (SD)	42.5 (43.0)	30.1 (22.9)	21.5 (11.3)	0.28

MICT, moderate-intensity continuous training; HIIT, high-intensity, low-volume interval training; PLA, sham placebo control group; H_b_A1c, glycosylated haemoglobin; NGSP, National Glycohemoglobin Standardization Program; VO_2peak_, peak oxygen consumption; AuC, area under the curve; FU, flux units; P%BL, peak as a percentage of baseline flux.

**Table 2 jcm-11-05018-t002:** Group mean change scores for primary and secondary outcomes.

	MICT (*n* = 10)			HIIT (*n* = 12)			PLA (*n* = 11)			
	Pre	Post	Change	Pre	Post	Change	Pre	Post	Change	*p* Value
AuC index	3.6 ± 2.3	4.1 ± 1.8	0.48 ± 1.52	3.8 ± 2.4	3.8 ± 1.9	−0.07 ± 1.72	3.1 ± 1.0	2.9 ± 1.0	−0.20 ± 0.82	0.25
P%BL	501.1 ± 200.0	504.7 ± 161.9	3.6 ± 161.6	542.3 ± 307.9	464.9 ± 269.6	−75.1 ± 388.0	447.9 ± 154.1	498.7 ± 228.0	50.8 ± 194.3	0.80
Peak	150.4 ± 117.2	150.3 ± 130.1	−0.1 ± 66.0	119.3 ± 79.2	157.2 ± 115.1	37.9 ± 132.8	152.9 ± 87.2	208.9 ± 151.0	56.0 ± 116.0	0.44
TtPeak	42.5 ± 43.0	23.4 ± 17.4	−19.1 ± 34.5	30.1 ± 22.9	31.0 ± 17.5	0.8 ± 34.9	21.5 ± 11.3	29.2 ± 18.9	7.7 ± 13.2	0.54
STP	102 ± 21.0	118.6 ± 20.5	16.3 ± 20.0	114.1 ± 20.0	116.7 ± 25.9	2.6 ± 28.9	104.4 ± 29.6	108.3 ± 33.8	3.9 ± 25.5	0.55
TBI	0.79 ± 0.15	0.88 ± 0.09	0.09 ± 0.16	0.80 ± 0.10	0.78 ± 0.12	−0.02 ± 0.17	0.76 ± 0.21	0.78 ± 0.18	0.01 ± 0.17	0.16
ABI	1.08 ± 0.06	1.05 ± 0.10	−0.04 ± 0.09	1.03 ± 0.13	1.09 ± 0.15	0.06 ± 0.17	1.06 ± 0.13	1.06 ± 0.14	−0.01 ± 0.08	0.42

MICT, moderate-intensity continuous training group; HIIT, high-intensity interval training group; PLA, placebo exercise group; AuC, area under the curve; P%BL, peak as a percentage of baseline flux; TtPeak, time to peak flux; STP, systolic toe pressure; TBI, toe-brachial index; ABI, ankle-brachial index.

**Table 3 jcm-11-05018-t003:** Group mean change scores for measures of body composition, cardiorespiratory fitness, biochemistry and lipids.

	MICT (*n* = 10)		HIIT (*n* = 12)		PLA (*n* = 11)		
	Baseline	Follow-Up	Baseline	Follow-Up	Baseline	Follow-Up	*p* Value
Body composition							
Body weight (Kg)	95.6 ± 16.9	95.3 ± 17.5	110.4 ± 14.5	110.2 ± 14.1	106.0 ± 16.9	107.3 ± 18.5	0.46
Body mass index (kg/m^2^)	33.9 ± 3.2	33.8 ± 3.4	37.5 ± 5.2	37.5 ± 5.1	35.8 ± 5.4	36.1 ± 5.8	0.54
Waist (cm)	110.2 ± 9.7	107.5 ± 9.2	122.0 ± 11.3	118.1 ± 12.5	116.2 ± 13.1	115.5 ± 12.3	0.11
Hip (cm)	118.4 ± 8.6	116.4 ± 9.6	123.4 ± 11.6	122.0 ± 9.9	118.7 ± 10.6	119.6 ± 11.1	0.24
Cardiorespiratory fitness							
Systolic brachial pressure (mmHg)	128.9 ± 11.4	133.1 ± 12.5	142.2 ± 9.8	138.5 ± 13.7	136.9 ± 18.6	142.7 ± 20.0	0.50
Heart rate (bpm)	69 ± 8	69 ± 5.9	68 ± 7.8	70 ± 6.5	74 ± 12.4	73 ± 13.3	0.84
VO_2peak_ (L/min)	2.1 ± 0.9	2.3 ± 0.7	2.3 ± 0.4	2.4 ± 0.4	2.0 ± 0.5	2.0 ± 0.6	0.40
VO_2peak_ (mL/kg/min)	21.2 ± 6.2	23.7 ± 7.5	20.8 ± 2.5	21.7 ± 2.3	19.5 ± 4.3	18.7 ± 4.7	0.05
Biochemistry and Lipids							
Aspartate transaminase (U/L)	23 ± 6.1	23.1 ± 4.6	39.1 ± 29.2	27.8 ± 16.5	25.6 ± 15.3	22.5 ± 7.1	0.77
Alanine aminotransferase (U/L)	32 ± 11.6	29.9 ± 12.5	44.3 ± 31.8	34.7 ± 28.5	29.8 ± 15.1	31.8 ± 17.1	0.38
C-reactive protein (mg/L)	3.7 ± 4.7	3.2 ± 2.5	4.5 ± 5.7	4.3 ± 3.5	4.3 ± 3.7	5.3 ± 4.9	0.42
Cholesterol (mmol/L)	4.5 ± 0.8	4.5 ± 0.5	4.5 ± 0.7	4.3 ± 0.9	4.5 ± 0.9	4.6 ± 1.0	0.67
Triglycerides (mmol/L)	1.6 ± 0.5	1.7 ± 0.4	1.5 ± 0.5	1.6 ± 0.6	5.3 ± 11.3	2.4 ± 1.4	0.28
High-density lipoprotein (mmol/L)	1.2 ± 0.3	1.2 ± 0.2	1.1 ± 0.2	1.1 ± 0.2	1.2 ± 0.3	1.2 ± 0.2	0.59
Low-density lipoprotein (mmol/L)	2.6 ± 0.7	2.5 ± 0.5	2.7 ± 0.6	2.5 ± 0.7	2.4 ± 0.7	2.4 ± 0.8	0.90
Fasting blood glucose (mmol/L)	7.7 ± 2.0	7.5 ± 2.3	7.6 ± 2.7	7.4 ± 1.5	9.1 ± 3.9	8.9 ± 3.6	0.69
Insulin (mU/L)	12.2 ± 5.5	12.4 ± 5.9	20.6 ± 30.3	34.8 ± 68.4	13.7 ± 6.1	32.0 ± 59.9	0.62
H_b_A1c—IFCC (mmol/mol)	53 ± 15	51 ± 11	54 ± 15	51 ± 8	60 ± 16	62 ± 15	0.07
H_b_A1c—NGSP (%)	7.0 ± 1.3	6.8 ± 1.0	7.1 ± 1.4	6.8 ± 0.7	7.6 ± 1.5	7.8 ± 1.4	0.07
Free fatty acids (μmol/L)	493.3 ± 225.1	364.4 ± 116.0	503.8 ± 196.8	488.4 ± 166.6	600.6 ± 168.7	473.0 ± 160.2	0.18

Presented as mean (±SD). MICT, moderate-intensity continuous training; HIIT, high-intensity interval training; PLA, placebo; VO_2peak_, peak oxygen consumption.

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
