# Peer review of "The Efficacy of Exercise Training for Cutaneous Microvascular Reactivity in the Foot in People with Diabetes and Obesity: Secondary Analyses from a Randomized Controlled Trial"

_jcm, 2022, doi:10.3390/jcm11175018_

Round 1
Reviewer 1 Report
This study by Lanting et al. investigated the effects of 12-weeks exercise training (either moderate-intensity continuous training or low-volume high-intensity interval training on cutaneous microvascular reactivity in the foot. The study was performed as secondary analysis of data from a randomized controlled study. Although some of the observations reported may be of interest, the trial was ended early and it is difficult to determine whether the negative results reported are due to the study being underpowered or due to a real lack of effect of exercise training. This limitation is recognized, but it is still unclear how this study adds to our existing knowledge in this area.
In addition to the considerable limitation of low power, the following questions also arose.
It does not appear that the volume of exercise (per week) was matched between HIIT and MICT. Was there a rationale for using these particular exercise interventions?
Baseline flux was not reported for any of the groups. This information should be included.
How was compliance determined? Was compliance an indicator of the number of sessions completed each week? It appears that compliance was much lower in the PLA group. Was there any reason for this?
How was VO2 peak determined? How was performance of 90% of VO2 peak for 4 minutes monitored in the individuals who performed HIIT? Please report heart rates during exercise bouts.
According to table 3, VO2 peak differed between groups or across time (p = 0.05). It does not seem that this was considered in the results or the discussion.
Author Response
- This study by Lanting et al. investigated the effects of 12-weeks exercise training (either moderate-intensity continuous training or low-volume high-intensity interval training on cutaneous microvascular reactivity in the foot. The study was performed as secondary analysis of data from a randomized controlled study. Although some of the observations reported may be of interest, the trial was ended early and it is difficult to determine whether the negative results reported are due to the study being underpowered or due to a real lack of effect of exercise training. This limitation is recognized, but it is still unclear how this study adds to our existing knowledge in this area.
RESPONSE: We would like to thank the reviewer for taking the time to review our manuscript and also for providing some useful considerations to enhance the quality of this article. Though underpowered, this study was conducted over several years, which highlights the difficulties in undertaking randomised control trials. Given that only two RCTs have investigated the effect of exercise interventions on cutaneous microvascular reactivity in people with diabetes (and none to date in lower limbs), further evidence is needed. We have provided the following to the first paragraph of the discussion (Lines 354-358) to highlight the value of this study:
“This is the first data from RCT providing lower limb cutaneous microvascular measures which is highly relevant given the overwhelming burden of diabetes-related foot disease. This data suggests that higher-volume of exercise and larger sample studies are important considerations for interventions involving physical activity.”
In addition to the considerable limitation of low power, the following questions also arose.
- It does not appear that the volume of exercise (per week) was matched between HIIT and MICT. Was there a rationale for using these particular exercise interventions?
RESPONSE: Thank you for your question. We do apologise if this was not clear. Both HIIT and MICT were conducted 3 times per week (Lines 200-202) for 12 weeks. The session duration did differ with MICT (40-55 min) and HIIT (19 min). Primarily, the low-volume HIIT protocol was implemented to gauge if this method resulted in health benefits that would thus provide a suitable option/benefit to people citing ‘lack of time’ or ‘boredom during exercise’ as barriers to physical activity. Please see line 79-80 of the introduction.
- Baseline flux was not reported for any of the groups. This information should be included.
RESPONSE: Thank you for your suggestion. Though not considered a primary outcome as it is not a measure of PORH, we agree that it warrants inclusion to provide further descriptive context. We have now reported baseline flux for all groups in Table 1 (Line 312).
- How was compliance determined? Was compliance an indicator of the number of sessions completed each week? It appears that compliance was much lower in the PLA group. Was there any reason for this?
RESPONSE: Thank you for these questions. Exercise session compliance was calculated as the total number of sessions attended/total number of available sessions × 100. We have changed the word ‘Adherence’ to compliance in the following sentence, for clarity (line 230-231):
“Compliance to the exercise interventions was calculated as a percentage of those completed from the number of sessions available.”
We have also addressed the lower poorer compliance in the PLA group (Lines 255-258):
“Reasons for poorer compliance in the PLA group were primarily due to the less frequent contact time or dissatisfaction with their group allocation”.
- How was VO2 peak determined? How was performance of 90% of VO2 peak for 4 minutes monitored in the individuals who performed HIIT? Please report heart rates during exercise bouts.
RESPONSE: Thank you for these questions. As reporting heart rate alone as a monitor for intensity is problematic given the “lag” in HR during HIIT [Buchheit et al., 2013 - DOI: 10.1007/s40279-013-0029-x], our prescription involved determining the linear relationship between workload in (watts) with oxygen consumption, determining the formula, then imputing the desired intensity (eg 50%, 60%, and 90% VO2peak). We have though, added the following to the manuscript (Lines 187-192):
“Measurement of cardiorespiratory fitness have been described previously [Way et al., 2020 & Sabag et al., 2020], though briefly, breath-by-breath analysis was collected (Ultima PFX pulmonary function/stress testing system, MGC Diagnostics) during a graded maximal exercise test to measure peak oxygen consumption (VO2 peak). All tests incorporated a 3-min warm-up at 35 W for women and 65 W for men. The workloads were incrementally adjusted by 25 W every 150 s until volitional fatigue”.
- According to table 3, VO2 peak differed between groups or across time (p = 0.05). It does not seem that this was considered in the results or the discussion.
RESPONSE: Thank you for your suggestion. The discussion is quite long currently, and we have needed to be more succinct to satisfy reviewers. However, we agree that this should be discussed at least briefly. We have thus added the following to the discussion (Line 442-443) so that it is at least provided to the reader:
“There was also a significant difference between groups for VO2peak (ml/kg/min) and this may have confounded findings”.
Reviewer 2 Report
I thank the authors and editor for providing me with the opportunity to review the manuscript entitled "The efficacy of exercise training for cutaneous microvascular reactivity in the foot in people with diabetes and obesity: secondary analyses from a randomized controlled trial," by Lanting et al. My primary suggestion is that the authors shorten the discussion in view of the study being underpowered. In it's current form, I find the discussion hard to follow.
Minor:
· Please define the abbreviation PORH on first use. I suspect Post-Occlusive Reactive Hyperaemia (PORH)?
· Should lines 185-186 be the inverse? i.e., BMI = weight (kg) divided by height (m) squared?
Major:
· While the authors acknowledge this in the limitations, the major concern is that the study is underpowered and is prone to type II error. I suggest the authors shorten the discussion and be more conservative when reporting the “no-difference” findings. Indeed, the authors have previously published a meta-analysis that suggests aerobic exercise is effective at improving cutaneous microvascular reactivity [1]. The manuscript is very long in general, and in view of the study being underpowered, I suggest making it more concise, especially the introduction and discussion.
-Results presented in Table 3. It doesn’t appear that the methodology for these outcomes are described and the results are not discussed. There also doesn’t seem to be a reference to Table 3 in the text. I suggest these results are removed if they are not discussed.
Author Response
- I thank the authors and editor for providing me with the opportunity to review the manuscript entitled "The efficacy of exercise training for cutaneous microvascular reactivity in the foot in people with diabetes and obesity: secondary analyses from a randomized controlled trial," by Lanting et al. My primary suggestion is that the authors shorten the discussion in view of the study being underpowered. In it's current form, I find the discussion hard to follow.
RESPONSE: We appreciate your time and experience in reviewing our manuscript as well as the suggestions you have provided to enhance its quality. We take your point regarding length of the discussion and as such we have removed significant sections of text (300 words). Please see the track changes in the document for context.
Minor:
- Please define the abbreviation PORH on first use. I suspect Post-Occlusive Reactive Hyperaemia (PORH)?
RESPONSE: Thank you for noticing this oversight. We have now defined PORH at first mention, (Line 108)
- Should lines 185-186 be the inverse? i.e., BMI = weight (kg) divided by height (m) squared?
RESPONSE: Thank you for noticing this error. This is now corrected, (Line 193).
Major:
- While the authors acknowledge this in the limitations, the major concern is that the study is underpowered and is prone to type II error. I suggest the authors shorten the discussion and be more conservative when reporting the “no-difference” findings. Indeed, the authors have previously published a meta-analysis that suggests aerobic exercise is effective at improving cutaneous microvascular reactivity [1]. The manuscript is very long in general, and in view of the study being underpowered, I suggest making it more concise, especially the introduction and discussion.
RESPONSE: Thank you for your suggestions for improving the readability of this manuscript. We agree that the discussion in particular was quite long, given the findings. Our previous meta-analysis suggesting that aerobic exercise improves cutaneous microvascular reactivity is generalisable to healthy older adults and thus detailed discussion was provided herein in the context of diabetes, where the effect of exercise is inconclusive. We have since now focused our discussion on the primary outcomes and deleted approximately 300 words. Please see manuscript for track changes and edits to the discussion section (As per your reviewer comment #1).
- Results presented in Table 3. It doesn’t appear that the methodology for these outcomes are described and the results are not discussed. There also doesn’t seem to be a reference to Table 3 in the text. I suggest these results are removed if they are not discussed.
RESPONSE: Thank you for this advice. The outcome measures and variables detailed in Table 3 are described in detail in previous publications and so the table was provided only to provide further context to the reader. We have however, added the following reference to table 3 in the results section (Line 257):
“Effects of exercise interventions on cardiovascular, metabolic and other outcome measures (Table 3) have been extensively described previously (Way et al., 2020; Sabag et al., 2020)”
Should the reviewers and editor wish us to remove table 3, still, we are happy to do so.
Reviewer 3 Report
Thank you for this insightful and well written paper. I have few minor revisions:
1. PORH is not preceded by proper elaboration (Page 3).
2. Compliance of PLA group was only about 50%. This should be mentioned as a major limitation.
Author Response
Thank you for this insightful and well written paper. I have few minor revisions:
RESPONSE: Thank you for taking the time to review our manuscript.
1. PORH is not preceded by proper elaboration (Page 3).
RESPONSE: Thank you for noticing this oversight. We have now defined PORH at first mention, (Line 108)
- Compliance of PLA group was only about 50%. This should be mentioned as a major limitation.
Thank you for suggesting this. We have addressed the lower poorer compliance in the PLA group (Lines 255-258):
“Reasons for poorer compliance in the PLA group were primarily due to the less frequent contact time or dissatisfaction with their group allocation”.
Furthermore in the limitations we have added (Lines 443-444):
“There was also a significant difference between groups for VO2peak (ml/kg/min) and this may have confounded findings, as may have the relatively low compliance of the PLA group”.
Round 2
Reviewer 1 Report
The limitation related to the power of the study should be stated clearly in the first paragraph of the discussion when the authors present the major negative findings of this study.
Author Response
Thank you for providing further review of our manuscript. We agree with your suggestion of highlighting the lack of power in the first paragraph of the discussion, and have done so accordingly (Line 354-355):
"The primary finding of this study is that in an adult population with type 2 diabetes and obesity, 12 weeks of supervised exercise (MICT or HIIT) did not have a statistically significant effect on PORH measured at the hallux, compared to sham exercise placebo, or to baseline measures (p>0.05). Though this study is underpowered, these are the first data from RCT providing lower limb cutaneous microvascular measures which, is highly relevant given the overwhelming burden of diabetes-related foot disease."
Reviewer 2 Report
I thank the authors for addressing my comments. While I am still unsure how much value the majority of table 3 adds, I am satisfied with the changes to the manuscript.
Author Response
Thank you for providing further review of our manuscript and for your consideration of our amendments.